# Dynamic Mechanical Properties and Microstructure of an (Al_0.5_CoCrFeNi)_0.95_Mo_0.025_C_0.025_ High Entropy Alloy

**DOI:** 10.3390/e21121154

**Published:** 2019-11-26

**Authors:** Bingfeng Wang, Chu Wang, Bin Liu, Xiaoyong Zhang

**Affiliations:** 1State Key Laboratory for Powder Metallurgy, Central South University, Changsha 410083, China; wangbingfeng@csu.edu.cn (B.W.); binliu@csu.edu.cn (B.L.); 2School of Materials Science and Engineering, Central South University, Changsha 410083, China; wangchu@csu.edu.cn

**Keywords:** high entropy alloy, mechanical properties, high strain rate, shear band, microstructure

## Abstract

The dynamic mechanical properties and microstructure of the (Al_0.5_CoCrFeNi)_0.95_Mo_0.025_C_0.025_ high entropy alloy (HEA) prepared by powder extrusion were investigated by a split Hopkinson pressure bar and electron probe microanalyzer and scanning electron microscope. The (Al_0.5_CoCrFeNi)_0.95_Mo_0.025_C_0.025_ HEA has a uniform face-centered cubic plus body-centered cubic solid solution structure and a fine grain-sized microstructure with a size of about 2 microns. The HEA possesses an excellent strain hardening rate and high strain rate sensitivity at a high strain rate. The Johnson–Cook plastic model was used to describe the dynamic flow behavior. Hat-shaped specimens with different nominal strain levels were used to investigate forced shear localization. After dynamic deformation, a thin and short shear band was generated in the designed shear zone and then the specimen quickly fractured along the shear band.

## 1. Introduction

High entropy alloys (HEAs) contain at least five elements in equiatomic or near-equiatomic ratios and have excellent properties [1,2,3,4]. They tended to form a simple solid-solution phase when first proposed by Yeh et al. [5]. With the further study of HEAs, multiphase HEAs became more and more common. As a broad category of HEAs, the Al_x_CoCrFeNi HEAs have aroused huge interest from researchers since the appearance of HEAs. The crystal structure of the Al_x_CoCrFeNi alloys transforms from face-centered cubic (FCC) to body-centered cubic (BCC) with an increase in Al content [6,7,8]. Meanwhile, the properties of HEAs, such as hardness, electrical conductivity, and thermal conductivity, also change with the transformation of the structure [6,7]. Lin et al. [9] investigated the microstructure, hardness, and corrosion properties of the Al_0.5_CoCrFeNi alloys aged at different temperatures and found that an FCC + BCC solid solution structure with optimal hardness appeared when the alloys were aged at 350–800 °C. Niu et al. [10] studied the strengthening of nanoprecipitations in an annealed Al_0.5_CoCrFeNi HEA and found that the nano-sized BCC phase particles reinforced the Al_0.5_CoCrFeNi HEA. Wang et al. [11] discovered that the temperature of the FCC to BCC phase transformation in the Al_0.5_CoCrFeNi HEA was 1044 K, and the tensile strength increased after phase transformation. Therefore, previous works showed that the existence of the FCC + BCC solid solution structure is beneficial for improving the mechanical properties of the Al_0.5_CoCrFeNi alloy. Regarding the effects of the addition of Mo and C elements to HEAs, the addition of Mo could cause solution strengthening because of the large atomic size [12,13,14,15]. Additionally, Zhuang et al. [16] found that the addition of Mo into Al_0.5_CoCrFeNi HEA could enhance the formation of a (Cr, Mo)-rich σ phase and adjust the mechanical properties of the Al_0.5_CoCrFeNiMo_x_ HEAs, including hardness, strength, and ductility. Jian et al. [17] suggested that the addition of C could promote the formation of carbides and induce precipitation strengthening. Jian et al. [17] also found that carbide precipitation could inhibit grain growth during recrystallization. The powder metallurgy method provides a promising way to prepare the Al_0.5_CoCrFeNi HEA reinforced by Mo and C elements.

Many materials researchers are paying more and more attention to the dynamic mechanical properties of HEAs for industry application. Kumar et al. [18] reached the conclusion that the deformation mechanism of the Al_0.1_CrFeCoNi HEA at a high strain rate is the same as that of low stacking fault energy materials. Dirras et al. [19] found that the yield strength in the Ti_20_Hf_20_Zr_20_Ta_20_Nb_20_ HEA was much higher under dynamic loading conditions than under quasi-static conditions and observed a shear band in the dynamic regime. Wang et al. [20] investigated the serration behavior and microstructure of the CoCrFeMnNi HEA prepared by powder metallurgy at strain rates from 1 × 10^−4^ s^−1^ to 1 × 10^−1^ s^−1^ and from 1 × 10^3^ s^−1^ to 3 × 10^3^ s^−1^. Additionally, the dynamic mechanical properties of the as-cast NiCrFeCoMn HEA were investigated by Wang et al. [21]. Ma et al. [22] studied the influence of the strain rate on the dynamic mechanical properties of the AlCrCuFeNi_2_ HEA and found a linear relationship between the dynamic yield strength and strain rate. Shear localization, or shear band, is the main failure mechanism for the materials deformed under high velocity loading. Li et al. [23] studied the shear localization in the CrMnFeCoNi HEA and revealed that the deformation mechanism was rotational dynamic recrystallization. They also found that the Al_0.3_CoCrFeNi HEA was unable to form a shear band [24], and they attributed the remarkable resistance to shear failure to the excellent strain hardening ability. An et al. [25] found that the addition of Mo and C elements to the Al_0.5_CoCrFeNi HEA could significantly increase the tensile strength under quasi-static loading. However, the dynamic mechanical properties of Al_0.5_CoCrFeNi reinforced by Mo and C are not clear yet.

In this paper, we investigated the microstructure and mechanical behavior of the (Al_0.5_CoCrFeNi)_0.95_Mo_0.025_C_0.025_ HEA prepared by powder extrusion and explored the forced shear localization of the (Al_0.5_CoCrFeNi)_0.95_Mo_0.025_C_0.025_ HEA.

## 2. Materials and Methods

High purity Al, Co, Cr, Fe, Ni, Mo, and C powders of a nominal composition (Al_0.5_CoCrFeNi)_0.95_Mo_0.025_C_0.025_ (at.%) were melted in an induction heated vacuum furnace. Then, the melt was drop through a ceramic tube and atomized by high purity Ar with a pressure of 4 MPa. After cooling down the gas-atomized powder in the atomization chamber, a stainless can with dimensions of φ 60 mm × 150 mm, which was degassed for 12 h at a temperature of 500 °C and sealed in a vacuum, was used to fill the as-prepared HEA powder. The encapsulated powder was then pre-heated for 1 h at a temperature of 1150 °C and immediately hot extruded into bars with an extrusion ratio of 6. After hot extrusion, the bar was cooled in the air. The content of oxygen in the gas-atomized powder was 750 ppm, and the chemical compositions of the (Al_0.5_CoCrFeNi)_0.95_Mo_0.025_C_0.025_ HEA are listed in Table 1.

For an n-element multicomponent alloy at a random state, the configurational entropy ΔS_mix_ can be calculated according to Boltzmann’s hypothesis by the following equation [26]:(1)$$\Delta S_{mix}=-R\sum_{i=1}^{n}\left(x_{i}\ln x_{i}\right)$$
where *R* is the gas constant and has a value of 8.314 JK^−1^ mol^−1^, *n* is the number of total elements, and *x_i_* is the concentration of element *i*.

By substituting the actual compositions of (Al_0.5_CoCrFeNi)_0.95_Mo_0.025_C_0.025_ HEA, the configurational entropy Δ*S_mix_* can be calculated as 1.725*R*, which satisfies the definition of HEA [27].

The produced material was observed using a POLYVAR-MET optical microscope (OM, Leica, Germany) and a JXA-8230 electron probe microanalyzer (EPMA, JEOL, Japan). The etchant for the (Al_0.5_CoCrFeNi)_0.95_Mo_0.025_C_0.025_ HEA was 25 mL hydrochloric acid + 25 mL ethyl alcohol + 5 g CuSO_4_·5H_2_O. An X-ray diffraction (XRD) test was also performed on a Rigaku 2550 diffractometer (Rigaku, Japan) using Cu radiation to analyze the structure of the as-received (Al_0.5_CoCrFeNi)_0.95_Mo_0.025_C_0.025_ HEA.

To obtain the mechanical properties of the (Al_0.5_CoCrFeNi)_0.95_Mo_0.025_C_0.025_ HEA, cylinder specimens were used for quasi-static compression tests and dynamic load tests. The cylinder specimens were cut from the as-received HEA bars along the extrusion direction (ED) with dimensions of φ 4 mm × 5.6 mm. The quasi-static compression tests were performed on INSTRON 8802 (Instron, the USA) with strain rates of 1 × 10^−1^, 1 × 10^−2^, and 1 × 10^−3^ s^−1^. The dynamic load tests were performed on a split Hopkinson pressure bar (SHPB, Beijing Xiaotu Technology Co., Ltd, China) with strain rates of 550, 1300, 2200, and 3000 s^−1^. Hat-shaped specimens cut along the ED were used to induce forced shear localization, as shown in Figure 1. The shadow parts show the designed shear zones of the specimen. Measured sizes of the hat-shaped specimens of designed shear zones are shown in the Table 2. The nominal strain was γ_nom._ = 2 × (f_before_ − f_after_)/(b_before_ − a_before_). After dynamic deformation, hat-shaped specimens were observed using a Quanta-200 scanning electron microscope (SEM, FEI, Netherlands).

## 3. Results and Discussion

### 3.1. Microstructure of the High Entropy Alloy

Figure 2a shows an optical micrograph of the (Al_0.5_CoCrFeNi)_0.95_Mo_0.025_C_0.025_ HEA. The HEA is composed of fine grains with a size of about 2 microns. Figure 2b shows the XRD pattern of the specimen. The phase composition of the (Al_0.5_CoCrFeNi)_0.95_Mo_0.025_C_0.025_ HEA includes the FFC phase and the BCC phase. The chemical compositions of the phases were identified by an electron probe technique, as shown in Figure 3 and Table 3. It was found that the (Al_0.5_CoCrFeNi)_0.95_Mo_0.025_C_0.025_ HEA consists of a light gray FCC matrix phase (marked as point 1), a black (Al, Ni)-rich BCC phase (marked as point 2), and a gray (Cr, Mo, C)-rich M_23_C_6_ carbide phase (marked as point 3). This shows that the C element exists in the M_23_C_6_ carbide phase and the Mo element exists in the FCC matrix phase and M_23_C_6_ phase, and the MoC phase can also be identified in the Mo-rich M_23_C_6_ carbide phase. Figure 3b displays that the distributions of the FCC, BCC, and M_23_C_6_ phases in the (Al_0.5_CoCrFeNi)_0.95_Mo_0.025_C_0.025_ HEA are uniform.

### 3.2. Stress–Strain Curves of the High Entropy Alloy

Figure 4a shows the engineering stress–strain curves of the (Al_0.5_CoCrFeNi)_0.95_Mo_0.025_C_0.025_ HEA tested at strain rates of 1 × 10^−1^, 1 × 10^−2^, and 1 × 10^−3^ s^−1^. The cylinder specimen exhibited yield strengths of 702, 703, and 716 MPa at strain rates of 1 × 10^−3^, 1 × 10^−2^, and 1 × 10^−1^ s^−1^, respectively. Therefore, with the increase of strain rates under quasi-static compression tests, the yield strength of the (Al_0.5_CoCrFeNi)_0.95_Mo_0.025_C_0.025_ HEA increased. It also can be observed from Figure 4b that light saw-like curves exist in quasi-static compression strain–stress curves.

Figure 5 shows the true stress–strain curves of the (Al_0.5_CoCrFeNi)_0.95_Mo_0.025_C_0.025_ HEA deformed at strain rates of 550, 1300, 2200, and 3000 s^−1^. The yield strengths of the (Al_0.5_CoCrFeNi)_0.95_Mo_0.025_C_0.025_ HEA were 754, 878, 958, and 1057 MPa at strain rates of 550, 1300, 2200 and 3000 s^−1^, respectively. The saw-like curves are much serious in dynamic compression strain–stress curves. As we observed from the optical micrograph and EPMA maps, the (Al_0.5_CoCrFeNi)_0.95_Mo_0.025_C_0.025_ HEA has small (Al,Ni)-rich BCC phase particles and M_23_C_6_ phase particles. The saw-like curves could be attributed to the interaction between the dislocation and (Al,Ni)-rich BCC phase particles plus the M_23_C_6_ phase particles.

The trend of the yield strength of the (Al_0.5_CoCrFeNi)_0.95_Mo_0.025_C_0.025_ HEA at quasi-static and dynamic loading tests is presented in Figure 6. The yield strength increased as a function of the strain rate, and the strain rate sensitivity *m* can be defined as follows:(2)$$m=\frac{d(log\sigma )}{d\left(log\dot{\varepsilon }\right)}$$
where σ and $\dot{\varepsilon }$ are the yield strength and strain rate, respectively.

The value of strain rate sensitivity under quasi-static testing conditions was 4.29 × 10^−3^, and the value of strain rate sensitivity under dynamic loading conditions was 0.192, which shows a strong strain-rate dependence of the (Al_0.5_CoCrFeNi)_0.95_Mo_0.025_C_0.025_ HEA at high strain rates.

The Johnson–Cook model, which is considered to have a huge strain, high strain rate, and high temperature, was used in this work to describe the plastic deformation of the (Al_0.5_CoCrFeNi)_0.95_Mo_0.025_C_0.025_ HEA. The equivalent flow stress σ can be expressed by the following equation:(3)$$\sigma =\left(A+B\varepsilon^{n}\right)\left(1+Cln{\dot{\varepsilon }}^{*}\right)$$
where A, B, and C are material constants, and ε and n are the equivalent plastic strain and the strain hardening exponent, respectively. ${\dot{\varepsilon }}^{*}$ is the normalized strain rate which can be expressed as ${\dot{\varepsilon }}^{*}=\dot{\varepsilon }/{\dot{\varepsilon }}_{0}$, where $\dot{\varepsilon }$ is the strain rate and ${\dot{\varepsilon }}_{0}$ is a reference strain rate that is equal to 1 × 10^−3^ s^−1^. Using the Matlab program (MathWorks, the US, Natick, Version 7.0) to fit the digital matrix of the stress–strain–strain rate, the parameter values and the Johnson–Cook physical equation for the (Al_0.5_CoCrFeNi)_0.95_Mo_0.025_C_0.025_ HEA were obtained. The Johnson–Cook model can describe the plastic deformation of the HEA at high strain rates well, as shown in Figure 7.
(4)$$\sigma =\left(397.1+1621\varepsilon^{0.45}\right)\left(1+0.05066ln{\dot{\varepsilon }}^{*}\right)$$

The strain hardening rate (dσ/dε) curve for the (Al_0.5_CoCrFeNi)_0.95_Mo_0.025_C_0.025_ HEA at the strain rate of 1 × 10^−3^ s^−1^ along with the stress–strain curve is shown in Figure 8a. It can be seen that the value of the strain hardening rate was 1621 MPa at a strain level of 0.2, which is much higher than that of most engineering structural alloys, including Al_x_CoCrFeNi HEA, as shown in Figure 8b. Therefore, the addition of Mo and C elements can improve the strain hardening rate of Al_x_CoCrFeNi HEA.

### 3.3. Shear Localization of the High Entropy Alloy

Figure 9 shows the hat-shaped specimens with different nominal strain levels used to investigate the forced shear localization of the (Al_0.5_CoCrFeNi)_0.95_Mo_0.025_C_0.025_ HEA. The hat-shaped specimen with a nominal strain of 2.53 was plastically deformed after dynamic deformation, and a small crack was generated at the upper part of the specimen, as shown in Figure 9a. When the designed nominal strain increased to 5.30, the designed shear zones in the hat-shaped specimen consisted of a predominant fracture zone and a small deformation zone, as shown in Figure 9b. Figure 9c is an enlarged view of the red rectangle in Figure 9b, and a short shear band with a width of about 10 microns can be observed. When the designed nominal strain increased to 9.83, the hat-shaped specimen was fractured, as shown in Figure 9d. Some dimples and streamlines were observed along the shear direction, as shown in Figure 9e, which mean that shear bands were generated and then the specimen rapidly fractured along the shear band.

The formation of the shear band can be attributed to the instability of thermal viscoplastic behavior. The critical condition of the constitutive instability can be expressed as
(5)$$\frac{d\tau }{d\gamma }=\frac{\partial \tau }{\partial \gamma }+\frac{\partial \tau }{\partial \overset{•}{\gamma }}\frac{d\overset{•}{\gamma }}{d\gamma }+\frac{\partial \tau }{\partial T}\frac{dT}{d\gamma }<0$$
where τ is the shear stress, γ is the shear strain, and $\dot{\gamma }$ is the shear strain rate.

The strain hardening ($\frac{\partial \tau }{\partial \gamma }$) and strain rate hardening ($\frac{\partial \tau }{\partial \dot{\gamma }}$) values are greater than 0, and the thermal softening ($\frac{\partial \tau }{\partial T}$) value is less than 0. When the value of thermal softening is greater than the values of strain hardening and strain rate hardening, shear localization happens. Li et al. [24] investigated the shear localization of Al_0.3_CoCrFeNi HEA and found the value of strain hardening was greater than the value of thermal softening, which explains the failure of the Al_0.3_CoCrFeNi HEA to form a shear band. In our work, with the addition of the Mo and C elements, the (Al_0.5_CoCrFeNi)_0.95_Mo_0.025_C_0.025_ HEA had a much higher strain hardening rate than that of the Al_0.3_CoCrFeNi HEA. However, the shear band was generated in the (Al_0.5_CoCrFeNi)_0.95_Mo_0.025_C_0.025_ HEA. This was due to the dissolved Mo atoms in the matrix phase, which hinder the heat diffusion, thus decreasing the thermal conductivity. The decreasing of thermal conductivity can increase the value of thermal softening, therefore promoting the generation of shear bands. Because of the high strain hardening rate and low interfacial strength [25], the shear band is very short and easily fractured along the shear direction. However, it shows that the shear band can coordinate deformation in the (Al_0.5_CoCrFeNi)_0.95_Mo_0.025_C_0.025_ HEA under high velocity loading.

## 4. Conclusions

The (Al_0.5_CoCrFeNi)_0.95_Mo_0.025_C_0.025_ HEA prepared by powder extrusion has a uniform FCC + BCC solid solution structure and consists of the FCC matrix phase, the (Al, Ni)-rich BCC phase, and the (Cr, Mo, C)-rich M_23_C_6_ carbide phase. The MoC phase can also be identified in the Mo-rich M_23_C_6_ carbide phase. The (Al_0.5_CoCrFeNi)_0.95_Mo_0.025_C_0.025_ HEA shows a strong strain rate sensitivity at high strain rates, and the yield strength of the HEA increases from 754 to 1057 MPa as the strain rate increases from 550 to 3000 s^−1^. The Johnson–Cook model was used to describe the dynamic flow behavior. The addition of Mo and C elements can improve the strain hardening rate of the Al_x_CoCrFeNi HEA. A short shear band of about 10 microns in width was generated in the designed shear zone and then the specimen quickly fractured along the shear band, which indicates that the shear band can coordinate deformation in the (Al_0.5_CoCrFeNi)_0.95_Mo_0.025_C_0.025_ HEA under high velocity loading.

## Figures and Tables

**Figure 1 entropy-21-01154-f001:**
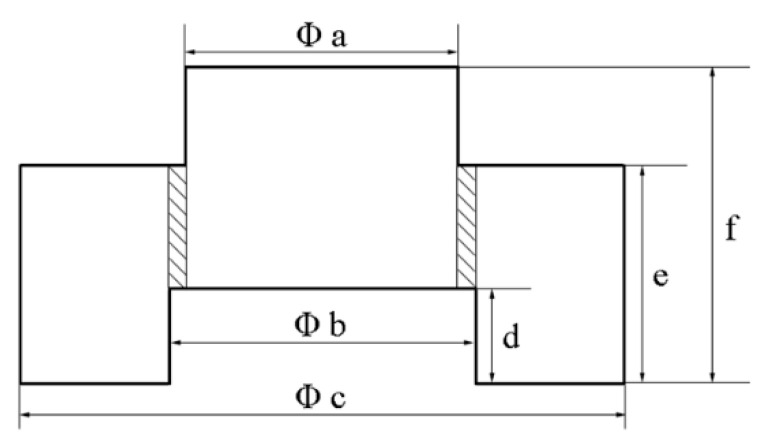
The diagram of the hat-shaped specimen.

**Figure 2 entropy-21-01154-f002:**
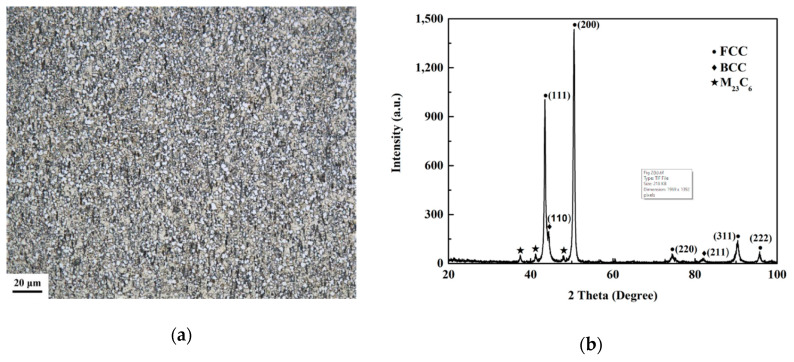
Microstructure of the (Al_0.5_CoCrFeNi)_0.95_Mo_0.025_C_0.025_ HEA. (**a**) Optical micrograph; (**b**) XRD pattern.

**Figure 3 entropy-21-01154-f003:**
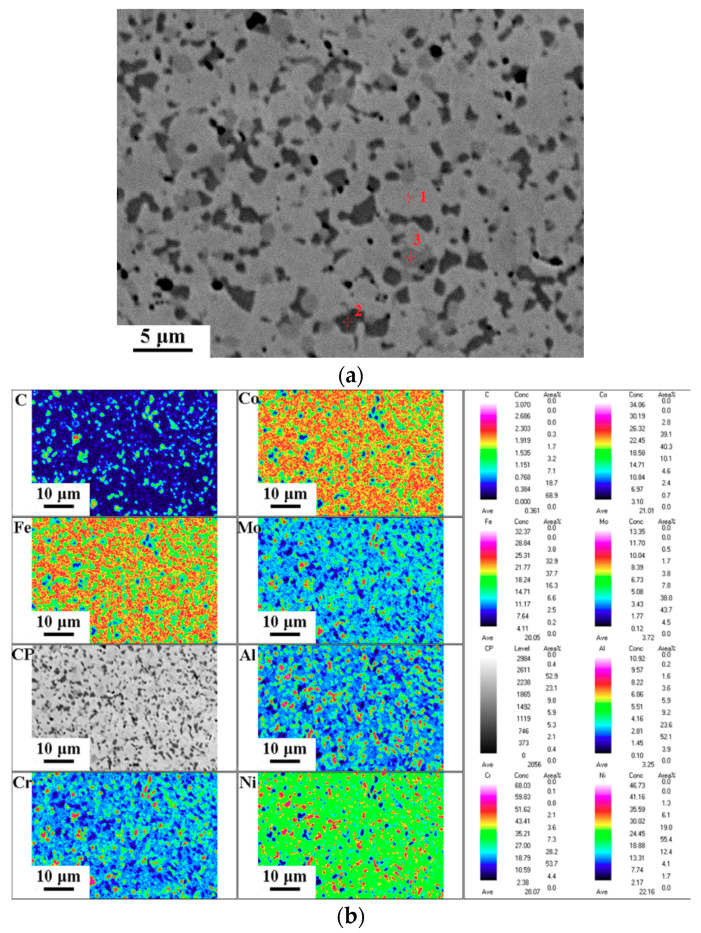
The electron probe microanalyzer (EPMA) maps of the (Al_0.5_CoCrFeNi)_0.95_Mo_0.025_C_0.025_ HEA. (**a**) The phase composition; (**b**) element distribution maps.

**Figure 4 entropy-21-01154-f004:**
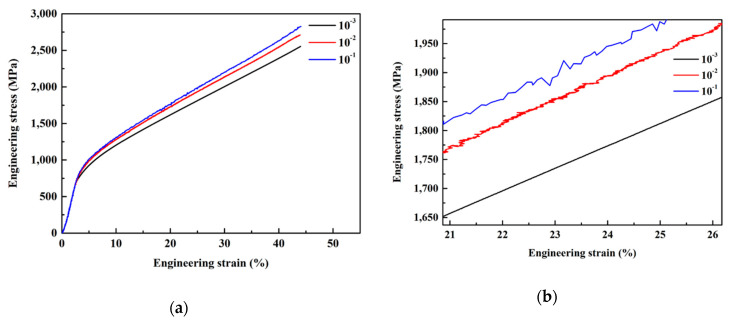
The engineering stress–strain curves of the (Al_0.5_CoCrFeNi)_0.95_Mo_0.025_C_0.025_ HEA tested at the strain rates of 1 × 10^−1^, 1 × 10^−2^, and 1 × 10^−3^ s^−1^; (**b**) the enlarged image of (**a**).

**Figure 5 entropy-21-01154-f005:**
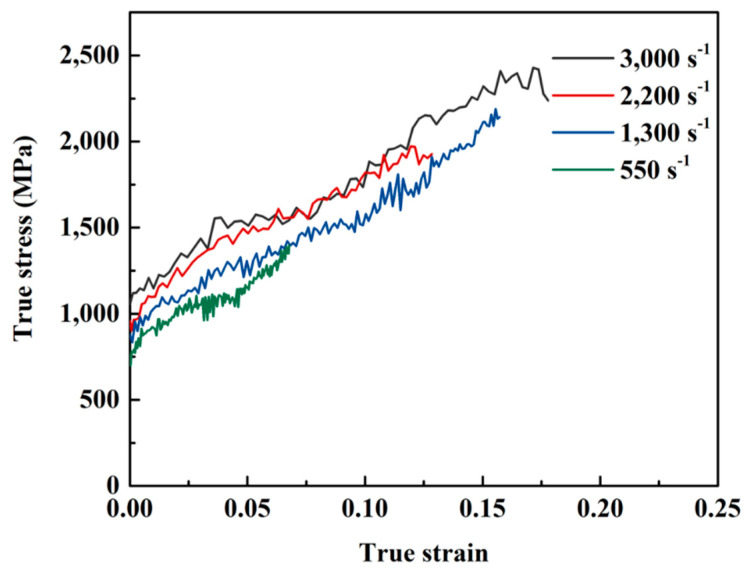
The true stress–strain curves of the (Al_0.5_CoCrFeNi)_0.95_Mo_0.025_C_0.025_ HEA tested at strain rates of 550, 1300, 2200 and 3000 s^−1^.

**Figure 6 entropy-21-01154-f006:**
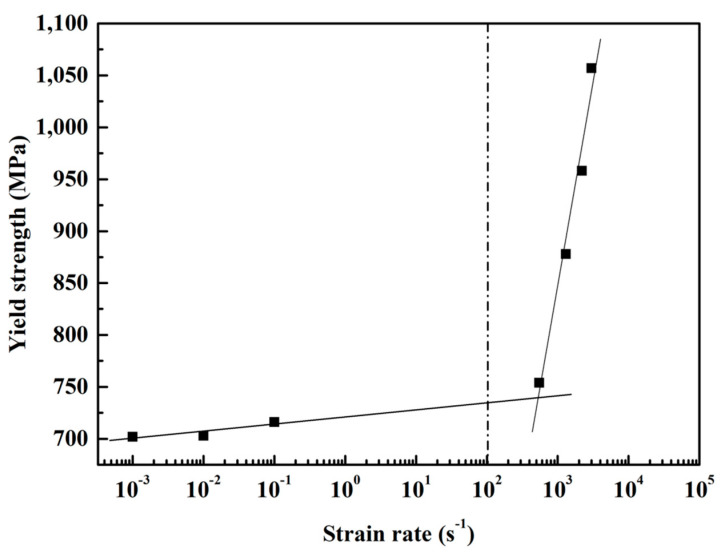
The strain rate sensitivity of the (Al_0.5_CoCrFeNi)_0.95_Mo_0.025_C_0.025_ HEA.

**Figure 7 entropy-21-01154-f007:**
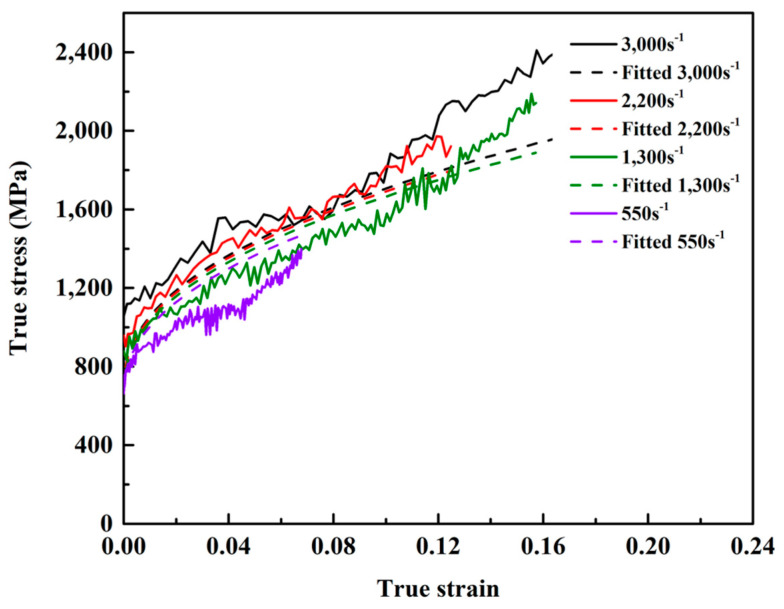
The curves calculated by the Johnson–Cook model compared with experimental curves at the strain rates of 550, 1300, 2200, and 3000 s^−1^.

**Figure 8 entropy-21-01154-f008:**
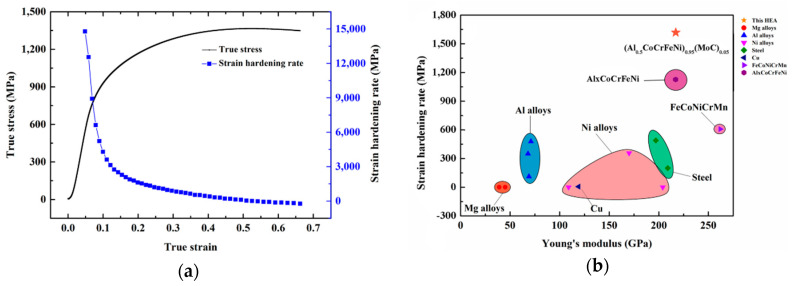
(**a**) The strain hardening rate (dσ/dε) and stress versus strain of the (Al_0.5_CoCrFeNi)_0.95_Mo_0.025_C_0.025_ HEA at a strain rate of 10^−3^ s^−1^; (**b**) the strain hardening rate of the (Al_0.5_CoCrFeNi)_0.95_Mo_0.025_C_0.025_ HEA compared with other engineering structural alloys.

**Figure 9 entropy-21-01154-f009:**
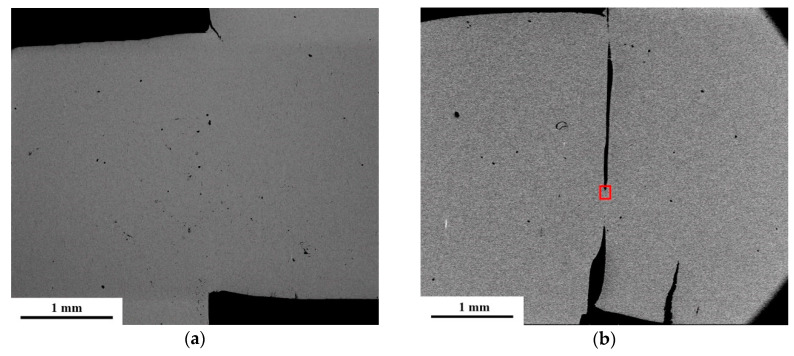
Microstructures of hat-shaped specimens with different nominal strains. (**a**) SEM graph for the specimen with a nominal strain of 2.53; (**b**) SEM graph for the specimen with a nominal strain of 5.30; (**c**) optical micrograph of the shear zone; (**d**) SEM graph for the specimen with a nominal strain of 9.83; (**e**) enlarged image of (**d**).

**Table 1 entropy-21-01154-t001:** The chemical compositions of the high entropy alloys (HEA).

Elements	Al	Co	Cr	Fe	Ni	Mo	C
(wt.%)	4.54	23.92	23.25	20.90	22.19	4.49	0.71
(at.%)	9.0	21.7	23.9	20.0	20.2	2.5	2.7

**Table 2 entropy-21-01154-t002:** Measured sizes of the hat-shaped specimens (dimensions in mm).

		a	b	c	d	e	f	γ_nom._
**1**	before	5.70	6.00	12.80	1.98	4.70	6.88	2.53
	after	5.96	6.00	12.84	1.78	4.64	6.50	
**2**	before	5.90	6.30	13.14	1.98	5.00	7.00	5.30
	after	6.20	6.58	13.26	1.00	4.98	5.94	
**3**	before	5.76	6.00	12.86	1.94	4.70	6.88	9.83
	after	5.92	6.00	12.90	0.90	4.64	5.70	

**Table 3 entropy-21-01154-t003:** The average chemical compositions of phases determined by EPMA (in at.%). BCC: body-centered cubic, FCC: face-centered cubic.

Phases	Al	Co	Cr	Fe	Ni	Mo	C
FCC	7.7	24.5	18.3	23.5	20.8	2.2	3.0
BCC	27.3	18.5	6.4	13.0	32.2	0.3	2.3
M_23_C_6_	1.1	6.7	55.9	9.4	3.2	5.8	17.9

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
