# Peer review of "Dynamic Mechanical Properties and Microstructure of an (Al0.5CoCrFeNi)0.95Mo0.025C0.025 High Entropy Alloy"

_entropy, 2019, doi:10.3390/e21121154_

Round 1
Reviewer 1 Report
The authors used powder metallurgy process to produce the experimental alloy, and studied the microstructure of the alloy, and the effect of strain rate on mechanical properties. Some questions are listed as below:
The overall compositions of the alloy should be listed in Table 2. The readers could understand the compositions changed during this process. Oxygen-content should also be listed in the Table, because the contamination of oxygen in powder metallurgy process is well known, and it strongly influence the mechanical properties. The authors should explain the name of the alloy, (Al5CoCrFeNi)0.95(MoC)0.05. No MoC (Mo-rich carbide) was found in this manuscript, only Cr-rich M23C6 was in the Table 2, Mo-content in this phase was only 5.8 at.%. Or the authors may change the name of alloy. The authors described “a refined grain-sized microstructure” in abstract. However, the authors did not show the origin morphologies and microstructures of the powders in this manuscript, so the authors should explain what the “refined” was. I did not see the reasons of forming of saw-like curves in Fig.4b and 5. The authors should explain that in the manuscript. XRD, Cu-Kα (Line 81)Author Response
Please see the attachment.

Reviewer 2 Report
Overall this is a excellent article. Regardless some entropy specific clarity is required as discussed below:
1) Please specify why (Al0.5CoCrFeNi)0.95(MoC)0.05 is actually a high entropy alloy. With what other alloy is the entropy of formation compared-please provide a numerical comparison? The reviewer understands the strain issues and configurational short range order issues. However there has to be a enthalpy (over the temperature range of formation) or entropy comparison to qualify the high entropy designation.
2) Although references by Wang et al [19] are discussed along with [1-3] vis. a vis. stacking fault energy - please provide evidence that this indeed is a feature of this alloy to call it a HEA.
3) There are enough contributions in quasicrystalline Al alloys that indicate a high strain rate rate exponent eq.1 leads to exceptional wear. These are quantified in articles that deal with Wear Rate of Quasicrystalline alloys compared to 2024 and 6061 multi-component alloys. Please check these references in J. of Mater Sci. Letts; Current Opinion in Chemical Engineering and other journals to compare the published exponents with this alloy(figure 4 and 5).
4) The authors have done a good job with equations 2, 3 and 4 and Figure 8. It is a good contribution from them.
5) The explanation and discussion in lines 183-196 is very high quality. However for this journal which is not a physical metallurgy journal, more entropic pathway explanations are required for relevance to the journal community.
Round 2
Reviewer 1 Report
I agree with this version of manuscript.